POINT OF VIEW

# A transatlantic perspective on 20 emerging issues in biological engineering

**Abstract** Advances in biological engineering are likely to have substantial impacts on global society. To explore these potential impacts we ran a horizon scanning exercise to capture a range of perspectives on the opportunities and risks presented by biological engineering. We first identified 70 potential issues, and then used an iterative process to prioritise 20 issues that we considered to be emerging, to have potential global impact, and to be relatively unknown outside the field of biological engineering. The issues identified may be of interest to researchers, businesses and policy makers in sectors such as health, energy, agriculture and the environment.

BONNIE C WINTLE*, CHRISTIAN R BOEHM*, CATHERINE RHODES, JENNIFER C MOLLOY, PIERS MILLETT, LAURA ADAM, RAINER BREITLING, ROB CARLSON, ROCCO CASAGRANDE, MALCOLM DANDO, ROBERT DOUBLEDAY, ERIC DREXLER, BRETT EDWARDS, TOM ELLIS, NICHOLAS G EVANS, RICHARD HAMMOND, JIM HASELOFF, LINDA KAHL, TODD KUIKEN, BENJAMIN R LICHMAN, COLETTE A MATTHEWMAN, JOHNATHAN A NAPIER, SEÁN S ÓHÉIGEARTAIGH, NICOLA J PATRON, EDWARD PERELLO, PHILIP SHAPIRA, JOYCE TAIT, ERIKO TAKANO AND WILLIAM J SUTHERLAND

**Competing interests:** The authors declare that no competing interests exist.

## Aims

Biological engineering is the application of ideas and techniques from engineering to biological systems, often with the goal of addressing 'real-world' problems. Recent advances in synthetic biology, notably in gene-editing techniques, have substantially increased our capabilities for biological engineering, as have advances in areas such as information technology and robotics. Keeping track of the challenges and opportunities created by such advances requires a systematic approach to gathering, assessing and prioritising them. Horizon scanning offers one way of filtering diverse sources of information to seek weak signals that, when contextualised, indicate an issue is emerging (*Amanatidou et al., 2012*; *Saritas and Smith, 2011*). Horizon scanning can also highlight a range of developments in their early stages, thus helping researchers, businesses and policy-makers to plan for the future.

Forward-looking exercises of this type bring together people from different fields to explore the possible implications of one field of study on another. For example, after identifying that very few conservation practitioners had even heard of synthetic biology in 2012, scientists from both disciplines convened in 2013 to explore how synthetic biology and conservation would shape the future of nature (*Redford et al., 2013*). In the same year, a horizon scan of emerging issues of interest to the conservation community (*Sutherland et al., 2014*) flagged the use of gene-editing to control invasive species or disease vectors. Since then, CRISPR/Cas9 approaches to controlling disease-carrying mosquitos (*Adelman and Tu, 2016*) and invasive species (*Esvelt et al., 2014*) have rapidly gained traction. This is not to suggest that such developments or applications are a product of being previously raised in horizon scanning activities, but that bringing an issue to the attention of the community early – before it becomes well known

– allows sufficient time to develop strategies for researching or managing the potential risks and opportunities accompanying these innovations.

As with any attempt to anticipate future trends, we acknowledge that the more speculative projections may not come to pass. Some technological hurdles may never be cleared, unexpected breakthroughs may change the direction of research, and some directions may be deemed too risky to pursue. We also recognise that providing a snapshot of such a broad range of issues comes at the expense of depth, so here we attempt only to provide a digestible summary and launching point for others to further explore those issues that may be relevant to them. For each issue outlined here, we aim to summarize possible implications for society, including questions, risks and opportunities.

How might an exercise such as this prove useful in the future? Outputs of similarly structured horizon scanning activities in Antarctic science (*Kennicutt et al., 2014*) have underpinned roadmaps outlining the enabling technologies, access to the region, logistics and infrastructure, and international cooperation (*Kennicutt et al., 2016*) required to "deliver the science". These have since been used to guide investment of national programs (*National Academies, 2015*). Similarly, the Natural Environment Research Council in the UK has drawn on annual horizon scans in conservation (see, for example, *Sutherland et al., 2014*) to inform their strategic planning. While a single horizon scan is only a first step in navigating the way forward (ideally, it would be followed with further exercises to map out how an agency might act in light of the information), we hope that the output of this scan may also be a useful starting point for developing policy designs.

Prioritising a set of issues for attention is an inherently subjective process, and reflects the perspectives and experiences of the people carrying out the assessment, as well as the dynamics of the group. This underscores the importance of bringing together a group that represents a wide range of perspectives. The main strength of this exercise is that the issues are systematically and democratically canvassed and prioritised by a relatively diverse group using structured elicitation and aggregation methods designed to mitigate some social psychological biases (*Burgman, 2015*), rather than reflecting the perspective of a single expert. Although we have attempted to capture an assortment of backgrounds, expertise, agendas and demographics (including age, gender and

career stage), we acknowledge that this article presents the perspectives of researchers based in the UK and US.

## Procedure

We followed a structured procedure developed by Sutherland et al. (*Sutherland et al., 2011*) to solicit, discuss and prioritise candidate issues (*Figure 1*). The method shares features of the Delphi technique (*Linstone and Turoff, 1975*), in that the scoring of issues is anonymous and iterative. It also draws on the collective wisdom of a group, while affording individuals opportunity to give private judgements, and to revise them in light of information and reasoning provided by others.

The horizon scan comprised a set of participants blending academic, industry, innovation, security, and policy expertise related to biological engineering, with a range of backgrounds in natural sciences, engineering, social sciences, and humanities. Each of the 27 participants (the authors minus facilitators) submitted short summaries of 2-5 'issues' that they considered to be on the horizon in biological engineering, and that have the potential to substantially impact global society. Participants also consulted their colleagues and networks for suggestions.

For submitted issues to be comparable with each other, they need to be framed at similar levels of granularity. Issues that are very broad, such as 'regulation of bioengineering', will encompass a whole suite of more detailed issues, so typically score higher than a single, highly specific issue. But these broad topics rarely make good horizon scanning issues, as they tend to be already well known, and are too vague to inform decision-making. To help ensure that issues were submitted with an appropriate level of granularity, an example topic was circulated that was framed at five different scales. The example was built around 'dual-use' scientific research (that is, research on materials or technologies that can be used to both benefit and harm humanity). As a general topic, dual-use research would be too broad for inclusion (Level 1). Likewise, a recent symposium hosted by the National Academies in the US (*National Academies, 2016a*) discussing a moratorium on 'gain of function' research (which is a type of dual-use research) was considered too narrow as a standalone issue (Level 5). A possible mid-point example (Level 3) would be the changing regulation of gain-of-function experiments, illustrated through the current US

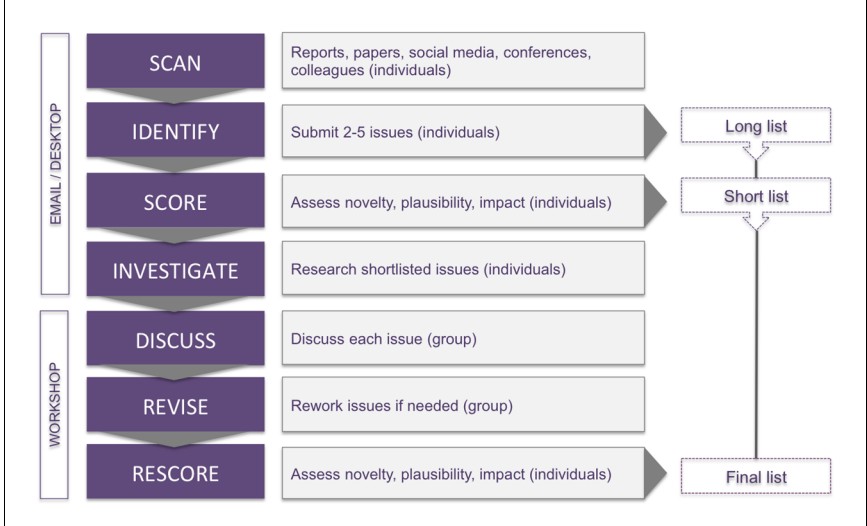

**Figure 1. Horizon scanning.** The seven stages of the horizon scanning procedure (*Sutherland et al., 2011*) used to identify emerging issues in biological engineering.
DOI: https://doi.org/10.7554/eLife.30247.002

moratorium on such research and related deliberative processes (such as the symposium hosted by the National Academies). Participants were asked to submit issues framed to approximate Level 3 granularity.

After merging duplicates, a total of 70 issue summaries were anonymized and circulated to all participants, who individually scored each issue according to its suitability (1-1000) as a horizon scanning issue. Suitability reflects a combination of plausibility, novelty, and potential impact on society in the medium to longer term future (up to 20 years, as a guide). Assessing potential impact on society is, of course, complex. Impacts might manifest via more direct or obvious effects, for example, on the environment or healthcare. But they may also arise indirectly, via impacts on funding, research, innovation and regulation of either the products or the practice of biological engineering. Those that profoundly influence the speed or direction in which biological engineering advances will, in turn, impact society. In their score sheets, participants also indicated whether they had already 'heard of' each issue.

The first round scores were converted to standardised Z scores. That is, the mean and standard deviation of each individual's set of scores were first calculated, then each item-score in the set was standardised by subtracting the mean and dividing by the standard deviation. The resulting Z scores retain information about the distribution of magnitudes in the

suitability scores and can be meaningfully aggregated across participants who have provided sets of scores with different means and variances. To explore divergence in scores across contributors, we analysed the inter-rater concordance of raw scores from Round 1 with Kendall's W, and the rank correlation between individual participants with Spearman's rho.

Issues in the original long list were ranked according to the average Z scores. Based on feedback during first-round scoring, one issue describing two distinct innovations was split into two, and two pairs of similar issues were each combined into one. Before finalising our short list of top-scoring issues, participants were given the opportunity to save issues that were about to be eliminated, if they wished to see them further discussed. Two issues were retained at this step, and one further issue was added later. A resulting short list of 34 remaining issues was taken forward for further discussion at the workshop.

Prior to the workshop, participants were each assigned 3–4 issues to investigate in more depth: this aim of this stage was to gather evidence to help assess whether the issue in question was sufficiently plausible, novel and consequential to warrant inclusion in the final list. Each of the shortlisted issues had 2–3 investigators assigned to it, who were generally not experts in that particular topic nor the person who submitted the issue. This meant that workshop discussion could include the person who submitted the topic, others who were already knowledgeable in the area, and the 2–3 people who had been assigned the topic, so allowing a more informed discussion.

In the workshop, convened in Cambridge, UK, in November 2016, participants systematically discussed each issue in turn. They were prompted to consider how well known the issue already was (based on the percentage of participants who had heard of it), what was novel about the issue, together with particular challenges and opportunities it presented. After working through each issue, participants individually and confidentially scored its suitability for a second time. At the end of the workshop, the 20 top-scoring issues (based on Z scores) were presented to participants for a final discussion. One issue was considered to be an example of another, and so was merged, allowing inclusion of the 21st-ranking issue in the final list. Another issue ('brain–machine interfaces') was considered to be outside the scope of 'biological engineering', so participants voted to swap it with a

slightly lower ranked issue. The final list of issues is reported below, roughly grouped according to their relevance in the near (< 5 years), intermediate (5-10 years), and longer (> 10 years) terms. 'Relevance' is a subjective and fuzzy measure and *could* be determined by any number of factors, such as whether the issue is considered underway, established, or at a tipping point by the indicated year. Applying criteria such as these is difficult for issues that are composed of multiple elements. So here, 'relevance' refers to how directly, and by extension, how soon the overall issue could measurably interact with risks to society. We endeavour to give a balanced view of both the risks and opportunities presented by each issue.

## Issues most relevant in the near term (< 5 years)

### Artificial photosynthesis and carbon capture for producing biofuels

There is a growing need to capture carbon and harness energy from sunlight in order to reduce the environmental impacts of fossil fuel combustion and methane release from large-animal agriculture. This would also enable production of fuels, plastics and chemicals from pollutants. Biology-based methods of carbon capture include using bioreactors to catalyse the production of fuels in fermentation bioreactors (Patent 20150247171, Lanzatech, NZ), or creating 'artificial photosynthesis' that uses solar energy to drive an electrochemical reduction of $CO_2$ to methanol (*Ager, 2016*). New research is focused on combining inorganic and biological systems to boost efficiency. For example, a 'bionic leaf' in which inorganic catalysts are interfaced with the bacterium *Ralstonia eutropha* is able to produce biomass and liquid fusel alcohols at carbon reduction efficiencies that exceed the rate achieved by photosynthesis in terrestrial plants (*Liu et al., 2016*). A hybrid nanowire-bacterial system for fixing $CO_2$ at high efficiencies has also been reported (*Liu et al., 2015*). Their hybrid approach, interfacing biological and inorganic systems, is scalable and might maximize the overall performance of chemical synthesis pathways. Such developments might contribute to the future adoption of carbon capture systems, and provide sustainable sources of commodity chemicals and fuel. However, the challenge of matching carbon flux to photon flux and managing both catalytic and biological components at large scale must still be overcome.

### Enhanced photosynthesis for agricultural productivity

Rapid population growth, accompanied by a changing climate, represents one of the major challenges of our time. In order to feed an expected world population of more than nine billion people in 2050, agricultural productivity will need to more than double in the face of shrinking croplands (*Alexandratos and Bruinsma, 2012*). Instead, growth in agricultural productivity of the most important food crops has been stagnating (*Ray et al., 2012*). The large gains in yield of the Green Revolution were primarily driven by increases in biomass partition into grain (harvest index), which is now near the theoretical upper limit. The so-called yield gap (the difference between possible and achieved yield) remains to be closed as well. To avoid further land conversion to farming, yields on currently farmed land need to be increased. Reducing pre-harvest losses, for example those caused by disease, as well as reducing post-harvest and post-consumer waste, will also help address food security. A promising approach to significantly boosting crop yields is to enhance photosynthesis. This has been discussed for some time, but synthetic biology is now providing the techniques to achieve it (*Furbank et al., 2015*). For example, an attempt is underway to increase yield potential by engineering a more efficient type of photosynthesis known as C4 into rice (http://photosynthome.irri.org/C4rice/). Models show that increased water and nitrogen use efficiencies from this engineering effort could result in yield increases of 30% to 50% (*Karki et al., 2013*). Synthetic biology techniques have enabled us to re-engineer entire microbial genomes (*Hutchison et al., 2016*), and efforts are underway to design synthetic chloroplast genomes in a similar manner (*Scharff and Bock, 2014*). In the future, engineered chloroplasts may encode functions for improved photosynthetic capture and conversion of light and carbon dioxide (*Ort et al., 2015*).

### New approaches to synthetic gene drives

The potential of gene drives (via the supra-Mendelian inheritance of an introduced trait) for modulating the insect vectors of human diseases such as malaria, West Nile Virus and Zika, has been widely recognised and much discussed (*National Academies, 2016b*). Gene drives are also being considered for restoring ecosystems by, for example, eliminating introduced predators from islands. They aim to increase the efficiency of existing 'non-drive' methods that

employ a similar concept of reducing population viability (e.g. by releasing male sterile insects into the environment), but that require repeat treatments. With the gene drive approach, the linked trait propagates to additional generations to more rapidly permeate the population, potentially spreading through the entire species. This raises questions about how deploying gene drives in wild populations might alter ecosystems, disrupting trophic levels and food webs, and creating vacant niches (for example, for new disease vector species or new disease organisms). Gene drives create risks that might be unpalatable even when balanced against the potential to reduce the number of lives lost to transmittable diseases. New innovations are therefore being developed to control their genetic reach (*Defense Advanced Research Projects Agency, 2016a*).

One approach is to include 'kill switches', such as identification markers or in-built susceptibilities to specific treatments or chemicals, which could be used to control the engineered population (*Akbari et al., 2015*; *DiCarlo et al., 2015*). A more complex proposition is to intrinsically self-limit the gene drive by deploying it as a 'daisy drive' in which the components of the gene drive are split into smaller, genetically-unlinked units that would eventually segregate in the population, inactivating the drive (*Smidler, 2016*).

Until gene drives have societal permission, their legal and regulated use will likely be restricted to proof-of-concept studies on confined laboratory populations and perhaps limited contained field trials. Given the current public and political debate, together with progress on alternative approaches to disease control (e.g. limiting the ability of mosquitos to transmit disease by release of *Wolbachia*-infected mosquitoes, release of genetically engineered sterile male insects, or the development of vaccines), it is still unclear whether gene drive techniques would become the technique of choice for disease control. The improvements in control of gene drives described here may increase the likelihood of such techniques being permitted, but uncertainty around the environmental impacts and the practical need for the technology may still render the risks unacceptable. For example, the Broad Institute prohibits its CRISPR technology to be used for gene drives, sterile seeds or tobacco products for human use (https://www.broadinstitute.org/news/licensing-crispr-agriculture-policy-considerations).

## Human genome editing

Genome editing technologies are accelerating our understanding of whole human genomes and individual genetic elements alike. Beyond basic research, many states have now taken steps to enable therapeutic genome editing in human somatic cells, and others have shown a willingness to directly modify human embryos for medical research (*Kang et al., 2016a*; *Liang et al., 2015*). Genome engineering technologies like CRISPR/Cas9 offer the possibility to improve human lifespans and health, recently being shown in human embryos to repair disease-causing mutations (*Ma et al., 2017*). However, their implementation poses major sociopolitical and ethical dilemmas. The question of inheritable, human germline editing grows increasingly relevant as more genome editing methods emerge (*Yang et al., 2014*), safe precedents are demonstrated in mammals, and somatic trials begin (*Reardon, 2016*).

Due to different levels of acceptance among individuals and worldviews, it is unlikely that there will be a universally agreed boundary between genome editing for preventative or therapeutic medicine and editing that aims for human genome perfection, or even enhancement. As knowledge about the genetics of increasingly subtle and complex human attributes accumulates, it is feasible that parents or states with the financial and technological means may elect to provide strategic advantages to future generations. For example, one Chinese leader previously stated that their government would use all means available to improve the health of the population, including direct genetic modification of its citizens (*Carlson, 2012*). With limited international discourse on individual and collective rights to genome editing, non-uniform use or regulation of the technology could transform social mobility and international order in unpredictable ways. As the technology advances, policymakers will need to work closely with regulators, biotechnology companies and healthcare providers to ensure that both somatic and germline human genome editing follows agreed ethical guidelines underpinned by extensive discourse.

## Accelerating defense agency research in biological engineering

The US Defense Advanced Research Projects Agency (DARPA) invested $110 million in synthetic biology in 2014, which accounted for almost 60% of funding for synthetic biology in the US that year, and this figure increases to

67% when other Department of Defense funding is included (*Kuiken, 2015*). The UK Defence Science and Technology Laboratory has also invested in synthetic biology, albeit on a smaller scale, and mainly focussed on developing novel materials (*DSTL, 2016*). Defense agencies report that they are investing in these programs with a view to preventing or responding to particular threats. However, areas in which some agencies are investing (e.g. agriculture, gene drives, chemical production) could raise both public perception issues and have dual-use potentials. For example, DARPA's Insect Allies Program intends to use insects to disseminate engineered plant viruses that confer traits to the target plants they feed on (*Defense Advanced Research Projects Agency, 2016b*), with the aim of protecting crops from potential plant pathogens. However, it is plausible that such technologies could be used by others to harm targets. Many ongoing military-funded bioengineering projects appear to focus on potential dual-use technologies (*Reardon, 2015*) and need to be carefully taken into account by regulators, as well as by funders, in order to avoid a security dilemma known as the 'spiral model'. This is where efforts to anticipate and counter adversary capabilities with engineered biological systems may actually produce those capabilities, justifying increased biodefense research and amplifying tensions (*Jervis, 1978*). Research programs will also need to be evaluated against various international agreements including the Biological and Toxin Weapons Convention, Chemical Weapons Convention, Convention on the Prohibition of Military or Any Other Hostile Use of Environmental Modification Techniques, and the Convention on Biological Diversity, which may require renegotiations to incorporate the rapidly changing technologies and proposed uses resulting from these programs.

## Issues most relevant in the intermediate term (5-10 years)

### Regenerative medicine: 3D printing body parts and tissue engineering

Tissue engineers have already built or grown transplantable bladders, hip joints, vaginas, windpipes, veins, arteries, ears, skin, the meniscus of the knee, and patches for damaged hearts (*Ghorbani et al., 2017*; *Davis et al., 2017*; *Naito et al., 2014*). Several scientific fields are coalescing to accelerate the development of methods to construct tissue. Would-be organ

engineers can now use custom-designed 3D printers to position cells accurately on organ-shaped scaffolds (*Wang et al., 2016a*), like parts in the chassis of a car. Complementing this technology, a technique known as whole-organ decellularization can create scaffolds ready for implanted cells while preserving the native tissue architecture (*Peloso et al., 2015*). Living cells printed on to structures have been implanted into animals and have matured into functional tissues (*Kang et al., 2016b*). As this technology advances, more ailments will be treatable, and, eventually, age-related degradation of various body systems may be reversible. While this technology will undoubtedly ease suffering caused by traumatic injuries and a myriad of illnesses that lead to organ failure, reversing the decay associated with age is fraught with ethical, social and economic concerns. Current healthcare systems would rapidly become overburdened by the cost of replenishing body parts of citizens as they age. If governments cannot afford costly therapies to ward off old age in all its citizens, new socioeconomic classes may emerge, as only those who can pay for such care themselves can extend their healthy years.

### Microbiome-based therapies

The human microbiome is implicated in a large number of human disorders, from Parkinson's to colon cancer (*Sampson et al., 2016*), as well as metabolic conditions such as obesity and type 2 diabetes (*Hartstra et al., 2015*). At present, interventions to manipulate the microbiome composition of humans are limited to rather crude approaches, such as probiotic and prebiotic diets and fecal transplants. However, synthetic biology approaches could greatly accelerate the development of more effective microbiota-based therapeutics (*Sheth et al., 2016*). For example, genetically engineered bacterial strains or consortia of natural and engineered microorganisms could be introduced to, or used to supplement, the host microbiome in cell-based therapies designed to prevent infection, resolve inflammation or treat metabolic disorders (*Mimee et al., 2016*). Engineered phage-based strategies may also prove useful in subtractive therapies aimed at targeting pathogens or shaping host-associated bacterial populations (*Citorik et al., 2014*). Among the regulatory challenges posed by these approaches is the possibility that DNA from genetically engineered microbes may spread to endogenous members of the microbiota through natural horizontal gene transfer, which is prevalent in the human

microbiome. Another concern is the unintentional colonization of others through escape of engineered organisms into the environment. A dialogue between researchers, clinicians and regulators to develop a coordinated regulatory framework for patient safety and environmental issues is needed to advance clinical research and translation of synthetic biology approaches to real-world microbiota-based therapies.

### Producing vaccines and human therapies in plants

Today, the majority of influenza vaccines are produced in embryonated chicken eggs in a six-month process, before which scientists must predict which strains will be dominant (*Milián and Kamen, 2015*). In 2014/2015, the vaccine was announced to be only 23% effective in the USA because the dominant virus had been incorrectly predicted and was not included in the vaccine development process (*Centers for Disease Control and Prevention et al., 2015*). The ability of plant platforms to rapidly respond to large-scale demand and emerging disease threats was first demonstrated in 2012, when DARPA issued a challenge to produce 10 million doses of the H1N1 flu vaccine within one month of receipt of an emailed genetic sequence. The Canadian company Medicago successfully responded using a wild-relative of tobacco for production (*Lomonossoff and D'Aoust, 2016*). The leaves of this cheap-to-grow plant are used in production runs of less than a week, requiring just water, light and the DNA template for the product of interest as inputs (*Sack et al., 2015*). Plants can now be tailored to produce proteins with human-like post-translational modifications as well as a range of other molecules used as human therapies that are prohibitively expensive or difficult to produce in other systems, widening the range of therapies that could be produced in plants (*Li et al., 2016*). The 2012 approval of Elelyso (Protalix) for commercial use in humans to treat Gaucher's disease has paved the way (*Mor, 2015*) for a number of therapeutics and vaccines targeting conditions ranging from influenza to non-Hodgkins lymphoma (*Holtz et al., 2015*). This widened scope and accumulation of examples and successes signals a shift toward rapidly deployed, industrial scale plant-based production of new therapies for emerging diseases, which will require an equally responsive regulatory landscape for testing and deployment.

### Manufacturing illegal drugs using engineered organisms

Advances in the engineering of microbial metabolism have led to the development of microbial strains capable of producing a wide range of complex molecules from sugar. These advances enable the fermentative production of drugs that would otherwise be produced chemically or isolated from wild-type organisms. A notable example of this is yeast engineered to produce opiates (*Galanie et al., 2015*). Although yields are currently insufficient for the isolation of significant quantities of products (*Endy et al., 2015*), it is anticipated that future advances may make fermentation not only a viable alternative to plant and chemistry based supply chains, but also attractive for criminal manufacture and abuse. The barrier to entry for fermentation is relatively low, and only a few cells are required to start a new culture. Thus the dissemination of engineered strains beyond academia and industry into groups operating outside state and international regulations could mark a sea change in both drug production and access (*Oye et al., 2015*). The unlicensed production of legal drugs by these means may result in cheaper, but possibly less pure, alternatives to licensed products (e.g. pharmaceuticals). Additionally, fermentative production of illicit drugs might enable small-scale, local manufacture that disrupts and undermines existing transit routes and organized crime networks. Or, technologies and individuals with appropriate expertise could possibly be incorporated into existing criminal networks. The potential for illegal use of these technologies will ensure that calls for their control and prohibition will continue (*Oye et al., 2015*).

### Reassigning codons as genetic firewalls

Whole-genome synthesis projects are underway in bacteria and yeast that may realise new, engineered microbes that only partially recognise the standard genetic code (*Ostrov et al., 2016*; *Wang et al., 2016b*). An *Escherichia coli* genome has been modified to no longer use one of the 64 codons normally recognised in protein synthesis: the cells containing this genome can instead use this free codon to programmably insert non-standard amino acids with alternative physical and chemical properties into proteins, while still translating the original protein repertoire required for growth (*Lajoie et al., 2013*). Codon reassignment offers attractive opportunities for industrial use, as cells can have

new chemistries added to their proteins and in doing so produce novel functional biomaterials or enzymes capable of new types of catalysis. It is currently aimed at 'stop' codons in the natural sequences, but could also be used to reassign amino acid-encoding alternative codons, in which case it could also create a genetic "firewall" where natural genes are no longer correctly converted into proteins when placed in the engineered cells. This reduces the susceptibility of the recoded organism to horizontal gene transfer from surrounding microbes or from attack by phage. While this is especially desirable for stability in industrial systems, it raises the possibility of creating invulnerable microbes that could grow unchecked in natural ecosystems. However, recoded cells can be limited to controlled conditions: by encoding the incorporation of a non-standard amino acid into essential genes, the cell is dependent on the supply of this amino acid for survival and will die in any environment where this is not provided (*Mandell et al., 2015*; *Rovner et al., 2015*). Recoded organisms thus present a new issue in biosafety: they have been intentionally designed to be less likely to interact with natural organisms but, in doing so, hold the potential to become an ecological competitor if not appropriately controlled. Genetically firewalled cells represent the gold standard for intrinsic containment, yet also a major challenge for existing regulation.

### Rise of automated tools for biological design, test and optimisation

The process of designing, testing and optimizing biological systems needs to become more efficient. Automation has been applied with great success to the design, test and manufacturing processes used in, for example, the automotive, aerospace and electronics industries. However, the engineering of organisms is not yet performed at similar scales. Automated fabrication with biological materials and the subsequent characterization of engineered materials and cells is now establishing itself in the form of services provided by 'biofactories' in a number of universities and companies (*Check Hayden, 2014*). Currently, the majority of laboratory automation is based around the use of existing tools and technologies, automating previously manually-executed protocols for design, simulation, building and testing. The next wave of lab automation will extend this, shifting experts' focus from the minutiae of organism design and construction to a more abstract functional view;

artificial intelligence-based software will automatically design and analyse experiments; and lab work will be performed by technicians or robots as instructed by the software. Hence, these tools make it possible to interrogate increasingly large experimental spaces rapidly and cheaply. This underlying technology will speed up the process to discover new molecules or prototype new applications fostering the development of many bio-based products. It will reduce the specialist skills needed for design, fabrication and validation and, along with outsourced fabrication, open up opportunities for countries with lesser biotechnology capabilities to take advantage of the booming bioeconomy. More broadly, the imminent arrival of "design for manufacturing" for bioengineered systems is likely to rapidly improve the ability of biomanufacturing to compete against traditional manufacturing industries (*Carlson, 2016*; *Sadowski et al., 2016*). The resulting acceleration of bioengineering will also impact the regulatory system as the complexity and rate of submissions rises.

### Biology as an information science: Impacts on global governance

The ability to chemically synthesise or 'write' DNA molecules at low cost means the inherent value of any given DNA sequence lies increasingly in information about its function or the function of any product it encodes rather than in a physical sample of the organism from which that sequence originated. Genetic information can now be accessed online and exploited in a remote location without engaging with complicated export/import procedures or material transfer agreements. While the use (and misuse) of genetic information historically required the transportation of specimens, today's biological engineers increasingly order the *de novo* synthesis of any DNA sequences that they wish to use from a commercial provider, using the sequence resources held in online databases as the template. Moreover, it is now possible to travel with a hand-held sequencer and to go from sample to sequence in less than 24 hours (*Quick et al., 2016*) negating the need to transport samples back to the laboratory to obtain the necessary genetic information. The enormous benefits of this rapid online transmission and synthesis of genetic information are already being realized, for example, through the production of ten million doses of vaccine just a month after receipt of an email containing the sequence of the viral strain (*Powell, 2015*). However, current

practices and guidelines for governing access, privacy and benefit sharing from the use of genetic resources, such as the Nagoya Protocol (see below), are still predominantly focused on physical samples, increasing the potential for biopiracy.

### Intersection of information security and bio-automation

Biological engineering puts genetic information at the heart of an iterative design-build-test cycle for genetically modified organisms. Advancements in automation technology combined with faster and more reliable engineering techniques have resulted in the emergence of robotic 'cloud labs' where digital information is transformed into DNA then expressed in some target organisms with very high-throughput and decreasing human oversight. This increased reliance on bio-automation and ingestion of digital information from multiple sources opens the possibility of new kinds of information security threats. These could include: tampering with digital DNA sequences leading to production of harmful organisms by researchers who are unaware of the malicious changes; sabotaging vaccine and drug production through attacks on critical DNA sequence databases or equipment; using DNA as a 'Trojan horse' to carry out a digital attack. The latter scenario was recently simulated by researchers from the University of Washington, who successfully engineered a DNA sequence to exploit a vulnerability they introduced into DNA sequencing software (*Ney, 2017*).

Information security is arguably a well-recognised threat so one might question why it is a horizon scanning issue. Emerging digital DNA tools and services present clear potential for new forms and sources of risk as DNA is directly 'executable' and verification methods such as sequencing can themselves be hacked, hampering efforts to assure quality and consistency. Recent experiences with 'internet-of-things devices' suggest that security does not always receive sufficient attention when a new technology is undergoing rapid development and increased decentralisation (*Department of Homeland Security, 2016*). Since bio-automation is currently undergoing such development and decentralisation, we propose that information security qualifies as an issue and that routes to tackling it should be explored as a priority. These might include setting information security standards for the bioindustry, such as ensuring strong encryption and quality control for all bio-

automation, recognizing public bioinformatics databases as critical infrastructure, and further engagement with information security experts when implementing tools and services.

### Effects of the Nagoya protocol on biological engineering

The Nagoya Protocol on Access to Genetic Resources and the Fair and Equitable Sharing of Benefits Arising from their Utilization (*Convention on Biological Diversity, 2010*), a supplementary agreement to the Convention on Biological Diversity (CBD), entered into force in 2014. It is expected to change the way genetic materials are treated by countries that are signatory to the protocol (93 at the time of writing). While many countries are still formulating national legislation and implementation plans, some countries rich in biodiversity (so-called provider countries) have already taken legislative steps to restrict access to physical and digital genetic resources originating from within their borders (*Bagley and Rai, 2013*; *Manheim, 2016*). Should the Nagoya Protocol be extended to associated data (such as genetic sequence information), it will substantially affect the collection, handling and transfer of such data which is used extensively in biological engineering. 'Digital Sequence Information of Genetic Resources' was discussed at a CBD meeting in December 2016 (the COP13 meeting) and recognised to be a cross-cutting issue (*Convention on Biological Diversity, 2016*). The decision was made to establish an Ad Hoc Technical Expert Group to compile relevant views and information, and a fact-finding and scoping study will likely be considered at the COP14 meeting. Regulatory uncertainty, restrictive terms set by provider countries, and limited capacity to deal with requests may slow down future research and its commercialization. In response, new programmes coordinating exchange of genetic resources may be implemented, potentially requiring an international system for tracking the origin of a genetic resource. Amidst its practical challenges, the developments discussed in 'Biology as an information science' (above) underscore the importance of maintaining the spirit of the Nagoya Protocol, for its potential to reduce inequality among countries and promote ecological sustainability by creating an incentive to preserve potentially valuable sources of genetic material.

### Corporate espionage and biocrime

Cutting-edge biotechnology is associated with a range of concerns about criminal misuse. Emergent bioeconomies will face many of the same hazards and vulnerabilities as more established sectors due to the high cost of biotechnology product development. The comparatively demanding regulatory environment in areas such as food and health may make the field particularly susceptible to both corporate espionage and the emergence of counterfeit markets. In a recent example, one of six Chinese nationals charged by the US Government pleaded guilty for attempting to steal trade secrets from GM seed companies (*Waltz, 2016*). Beyond theft of physical samples, the information-centric character of modern biotechnology entails increased risk of cybercrimes such as data theft and extortion (*Evans and Selgelid, 2015*). Underground trading already exists for recreational drugs, medicines, and crop seeds (*Tatge, 2004*). With continuing expansion of the biotech industry and increasing accessibility of both biological information and genetic engineering techniques to non-specialists, bio-piracy will likely become more widespread in the future.

## Issues most relevant in the longer term (>10 years)

### New makers disrupt pharmaceutical markets

Currently, many medicinal compounds are either chemically synthesized or extracted directly from the source organism, often a plant that is difficult to cultivate. These processes can be complex and costly, requiring specialized facilities. Recent advances have seen biosynthetic pathways for several human therapies re-engineered into yeast (for example, the analgesic hydrocodone (*Galanie et al., 2015*), the anti-malarial artemisinin (*Paddon et al., 2013*), and strictosidine, from which the chemotherapeutic agents vinblastine and vincristine are derived (*Brown et al., 2015*)). Additionally, community bio-labs and entrepreneurial start-ups around the world are customizing and diffusing methods and tools for biological experiments and engineering. For example, in 2015, a biohacking team in Oakland, California, secured crowdfunding to develop an open source protocol for making inexpensive generic insulin from *E. coli* (*Stelzer, 2016*). Alternative low-cost production systems combined with open business models and open source technologies herald opportunities for the distributed manufacturing of therapies tailored to regional diseases that multinational pharmaceutical companies might not find profitable (*Pauwels, 2016*). This could result in a shift toward more equitable and globally distributed pharmaceutical production, addressing current long-standing concerns that the pharmaceutical industry is profiteering from genetic samples taken from developing countries without sharing benefits (*World Health Organization, 2007*). However, it raises concerns around the potential disruption of existing manufacturing markets and raw material supply chains as well as fears about inadequate regulation, less rigorous product quality control, and misuse.

### Platform technologies to address emerging disease pandemics

Emerging infectious diseases – such as the recent Ebola and Zika virus outbreaks – and potential biological weapons attacks require scalable, flexible diagnosis and treatment (*World Health Organization, 2016*). Current methods of diagnosing and responding to disease tend to be tailored to individual pathogens, or even individual strains of pathogens, with little capacity to share data or reuse systems for multiple pathogens. These 'stovepipe' methods of engineering (referring to separate, isolated approaches to solving problems) are often inefficient when compared to reusable platforms that can adapt to detect and rapidly develop countermeasures to different emerging infectious diseases. Such platform technologies would greatly decrease response time to emerging pandemics. As such, there have been a series of recent funding calls for such platforms, for example, by the World Health Organization (*World Health Organization, 2016*). Platform technologies could use metagenomic sequencing to create pathogen-blind diagnoses, or be capable of creating a range of therapeutic agents. Existing examples include standardized influenza vaccine backbones for the rapid development of vaccine candidates (*Dormitzer et al., 2013*), and plant-based antibody production systems (*Olinger et al., 2012*), as described in 'Producing vaccines and human therapies in plants' above.

The value, distribution, and use of a particular platform technology is not guaranteed. Novel technologies to combat infectious disease are insufficient solutions if a significant portion of the population has no access to the most basic public health and healthcare infrastructure (*Evans, 2014*). Platform technologies

may or may not be fully distributed, and there may be restrictions on where they can operate; for example, if plant-based production systems have a limited climate in which they can be grown, or require significant resources to cultivate. Given that global protection against emerging infectious disease hinges on rapid, often international action, the political and economic barriers to the distribution of such technologies need to be addressed (*Brown and Evans, 2017*).

## Challenges to taxonomy-based descriptions and management of biological risk

Today, efforts to describe and manage biorisk are based upon taxonomic classification of the agents involved (for example, the Australia Group's 'Lists of pathogens' or the CDC's 'Select Agent Rules'). As the life sciences advance, the utility of these lists is diminished due to several factors. To begin with, chimeric and modified agents do not fit easily into such lists. For example, a virus composed of genetic elements from several related strains defies taxonomic description (e.g. is it a strain of measles virus, rinderpest virus or canine distemper virus?) and challenges safe handling guidelines. Secondly, and perhaps most fundamentally, it is the presence of particular functional properties of an agent that drive the risk, rather than the identity of the agent itself. For example, most strains of *Bacillus cereus* are harmless, but the identification of toxigenic strains (*Okinaka et al., 2006*) has prompted the US Centers for Disease Control and Prevention (CDC) to rewrite rules and include this strain as a select agent (*US Department of Health and Human Services, 2016*). Likewise, they have had to include at least one phenotypic definition, for Newcastle Disease Virus. The feasibility of describing and managing biorisk according to biological function, rather than taxonomy, has been the subject of a long-running debate (*National Academies, 2010*), but is extremely relevant at a time when many new pathogens and strains are being discovered through bioprospecting, and the current system risks overregulating harmless non-pathogenic organisms or failing to capture distantly related pathogens with similar properties. To ensure that biorisk management and biosecurity regimes remain relevant into the future, taxonomic lists would benefit from supplementary phenotypic definitions that capture the traits that influence the strains' biosafety or biosecurity risk.

## Shifting ownership models in biotechnology

Models of ownership in bioengineering are typically strongly vertically integrated and rely heavily on the patenting of both tools and applications. The current market structure and supply chain provides little access to basic bioengineering tools and technologies to those in low-resource settings who could arguably reap the greatest social and economic benefit from developing a sustainable bioeconomy based on local needs and priorities (*Juma and Konde, 2013*). The rise of off-patent, generic tools and the lowering of technical barriers for engineering biology has the potential to change this, particularly where new foundational advances are made open for others to build on (*Hope, 2008*). This is demonstrated in open source software and, more relevantly, in drug discovery (*Masum, 2011*). Current examples in biotechnology include the work of New Harvest, a US non-profit organisation that is building a library of open source cell lines for cultured meat production, and numerous open source providers of open hardware that enable high-throughput experimentation, such as the OpenTrons liquid handling robot and DropBot digital microfluidics system. Although platforms such as espacenet.com and lens.org help promote transparency, the patent landscape for engineering biology is complex (*Carbonell et al., 2016*; *Carlson, 2011*). Publicly available resources clarifying the status of open source biotechnologies could provide great benefit for enhancing the public's return on investment in research and in the patent system itself. Leveraging open source biotechnologies – those that entered the public domain via the patent system as well as those made available through legal tools such as the BioBrick Public Agreement and OpenMTA – could facilitate widespread sharing of knowledge and foundational tools for engineering biology (*Grewal, 2017*).

## Securing the critical infrastructure needed to deliver the bioeconomy

Many governments see a thriving bioeconomy as the basis of national prosperity in the 21st century, and synthetic biology will be a key component of the infrastructure needed to deliver this (*Carlson, 2016*). The UK Synthetic Biology Leadership Council (SBLC) Strategic Plan (*Synthetic Biology Leadership Council, 2016*) focuses attention on the translation of emerging ideas and commercialisation of applications with

the target of a £10 billion synthetic biology based platform in the UK by 2030, building on maximizing the capability of the innovation pipeline, building an expert workforce, developing a supportive business environment, and building value from national and international partnerships. A diverse, widely distributed and varied infrastructure will be critical, in the sense of being essential to the delivery of the expected benefits from the bioeconomy, and as such worthy of protection. However, the bioeconomy will subsume many sectors in a nation's economy (pharmaceuticals and health care; energy and transport; agriculture, food and fibre production; water and waste management; and potentially electricity generation). Vulnerability to criminal or terrorist activity is a legitimate concern, but the widely dispersed nature of this infrastructure (geographically and sectorally) will put it in a different category from what are currently considered National Critical Infrastructures (*Lewis et al., 2013*). More damage could therefore be caused to the bioeconomy by well-meaning attempts to protect it from threats than by the threats themselves. Countries are concerned that any loss of competitive edge would seriously impact their national security, via economic opportunity costs and the impeded development of specific security applications, such as developing medical countermeasures to threats and improving diagnostics (*Gronvall, 2015*). Picking the appropriate governance modalities will require balancing the freedom to innovate against the security benefits of centralisation and control (*International Risk Governance Council, 2011*).

## Discussion

Having completed the iterative process of culling low-scoring issues as described above, we found little separating the top 20 issues. To avoid over-emphasizing slight differences in score, we chose not to present the top 20 issues in rank order. The complexity of the field requires a comprehensive, multi-faceted approach. Still, some policy- or decision-making bodies may focus on preparing for distant futures, and on long-term issues that may otherwise be overshadowed by current, more pressing priorities. Others may focus on nearer term issues that require immediate attention. Large science funding bodies tend to consider a diverse suite of issues spread across a range of time horizons. To put the top 20 issues in a temporal context, we roughly grouped them

according to their relevance in the near (< 5 years), intermediate (5-10 years), and longer (> 10 years) term. Applications and research with the potential for near-term impacts on critical systems, such as global food and fuel supply, ecosystems, health and geopolitics, thus appeared in the first category. Those that influence society indirectly via platforms, ownership models, markets or future infrastructure, may have less immediate societal impact and thus appeared in the latter category.

The top 20 items contain a mixture of social, environmental, economic and health issues, but the balance was weighted towards medical innovations and health, with comparatively less representation of environmental, ecological or agricultural issues (with the exception of three topics: ecological risks of gene drives; artificial photosynthesis for producing fuel; and new enhanced photosynthesis methods for improving agricultural productivity). Surprisingly, this imbalance does *not* reflect the group's prevailing areas of application, which were more representative of food, agriculture and environment than health and medicine. This outcome may reflect the relatively high importance society places on human health and enhancement, and the far higher level of investment in health sciences. But just as biological engineering is poised to transform healthcare, it is also appears set to revolutionise these other fields. In the latest horizon scan for issues likely to impact the future of global conservation and the environment (*Sutherland et al., 2017*), four of the 15 prioritised topics were biotechnology applications: creating fuel from bionic leaves; reverse photosynthesis for biofuel production; manipulating coral symbionts to avoid mass coral bleaching; and extensive use of bacteria and fungi to manage agricultural pests and diseases.

### *Emerging themes*

Bioproduction and its intersection with the informational and digital aspects of biotechnology featured heavily in our issues. We raised the issue of increasingly distributed manufacturing on pharmaceutical markets, and much discussion was devoted to security of outsourced biomanufacturing, an area flagged as needing more research and policy, and one that is relatively underrepresented in the literature. The growth of the bio-based economy promises sustainability and new methods for addressing global environmental and societal challenges. Yet, at the same time, some aspects of the operation of the bioeconomy present new kinds of security

challenges. It is not only less centralised than more established industries, such as petrochemicals, but biological output may also present more complex, unknown, and large scale hazards than a vat of chemicals: in part because it is self-replicating and a significant proportion of its instruction set is digitally encoded in a readable-writable state. We flagged some challenges this creates for international agreements such as the Nagoya Protocol (e.g. controls on physical materials may be circumvented by synthesising organisms based on transmission of data instead). We discussed how this interdependency with information technology has also set the stage for new biothreats, with increasing opportunities to tamper with bio-data, algorithms or automated biofabrication systems. Biological data is distinct from other cyber security issues because we are inextricably intertwined with it; you can easily change your PIN or phone number, but it is not so easy to change your DNA. Standardising biological information and methods for validating, storing and retrieving data is seen as a starting point for improving cyber biosecurity, and efforts are being made to bring standardization into the field through national agencies (see, for example, *National Institute of Standards and Technology, 2014*; *British Standards Institution, 2015*) and community initiatives such as the international genetically engineered machine (iGEM) competition for students, which has led the way in standardisation of biological parts and descriptions.

Another theme that repeatedly emerged in our discussion was that of access to the technology. Issues around (in)equality were captured in several of the issues described in this paper. For example, the rise of open source, off-patent tools could facilitate widespread sharing of knowledge within the biological engineering field and increase access to benefits for those in developing countries. Translating increased equity in knowledge exchange and ownership into economic and sustainable development is subject to overcoming many existing inequalities and power structures, but some initiatives are beginning to bridge this gap, particularly in healthcare. Society may see the benefits of affordable medicine as new makers enter healthcare, reducing the monopolies of large, developed-world pharmaceutical companies, mediated through patents. On the other hand, some advances in the field may introduce *less* affordable, specialised healthcare as we move towards regenerative medicine – 3D printing of body parts, tissue engineering, and genetic upgrades – and augmenting human genomes, raising the possibility of new 'sociogenetic' classes.

A third theme extends from the discussions around equality, access and benefit sharing: that is, public trust and acceptance. A number of issues were discussed that might influence public acceptance of biotechnology in various ways. Acceptance may increase with the shift in ownership models described above – away from big business and towards more open science – and a more equal distribution of benefits. It may also increase as technologies advance to target problems that disproportionately affect the developing world, such as food security and disease. If synthetic biology were to be successful in eradicating malaria or Zika, this could bolster public opinion in favour of genetic engineering (as evidenced by the recent open letter of Nobel laureates criticising Greenpeace over its anti-GMO stance; *Agre, 2016*). Nonetheless, we note that in a recent vote in Florida – a non-binding referendum asking residents of Monroe county and Key West whether they support the release of genetically engineered mosquitoes to combat the spread of certain mosquito borne diseases – only a small majority of voters across the county supported the use of the technology (57%), and in the proposed field trial site, a majority opposed it (65%; *Servick, 2016*). Having an epidemiological end point as a measure of success of the technology could potentially mark a paradigm shift in the field, beyond the public acceptance yardstick. But then again, proceeding without the appropriate safety precautions and societal consensus – whatever the public health benefits – could damage the field for many years to come.

### Regulatory context

Emerging regulatory challenges that were raised in the longer list of horizon scanning issues (but not covered in the 20 issues above) include questions around the status of innovative products and processes given existing EU regulatory systems as applied to GMOs, regarded by many as 'not fit for purpose' (*Baulcombe, 2014*) (although the EU Opinions on Synthetic Biology disagreed with this assessment (*Scientific Committee on Emerging and Newly Identified Health Risks (SCENIHR), 2017*). The current EU regulatory system, if applied without adaptation to synthetic biology and gene editing techniques, may inhibit the development of innovations with the potential to deliver societal

benefits (*Tait, 2009*), such as the Arsenic Biosensor Collaboration for detecting unsafe arsenic levels in water wells in affected countries like India and Bangladesh. The current US regulatory system has also seen some challenges regarding the regulatory route to market for new products. Some crops engineered using programmable nucleases including CRISPR/Cas9 have fallen outside the mechanism used by the US Department of Agriculture (USDA) to capture a GMO product within its regulatory system because they contain small deletions rather than foreign genetic materials; e.g. non-browning potatoes and mushrooms (*Clasen et al., 2016*; *Yang, 2015*). Such cases have contributed to pressures for regulatory systems to be based on a risk−benefit analysis of the final product rather than the technology used to achieve it (*Camacho et al., 2014*).

A broader challenge, also raised in 'Securing the critical infrastructure needed to deliver the bioeconomy' above, lies in achieving the balance between regulation and opportunity costs – caution is necessary to ensure developments are safe and beneficial, but the regulatory approach to delivering such safeguards needs to be proportionate to the relevant costs and benefits. Risks around environmental impact, potential for weaponization, and narcotic production have prompted some groups to push for a moratorium on some of these technologies (*ETC Group, 2017*). If calls to ban certain biotechnologies are successful, or effective risk mitigation strategies are not in place before an accidental or deliberate adverse event occurs, we may see policy responses (*Morse, 2014*) that impede the delivery of potential benefits. While none of these tensions are new, the way they play out will fundamentally influence the future direction of biological engineering, including the issues we outline in this paper.

There is a general awareness of the need for regulatory reform. In July 2015, the White House issued a memorandum directing the three agencies responsible for overseeing biotechnology products in the US – the Environmental Protection Agency (EPA), the Food and Drug Administration (FDA), and USDA – to update the existing regulatory framework and "develop a long-term strategy to ensure that the system is prepared for the future products of biotechnology" (*Holdren, 2015*). In the EU, there is a different set of concerns about the operation of regulatory systems, and government policies focus increasingly on the need for more proportionate and adaptive regulatory systems

(*Tait and Banda, 2016*). A range of regulatory adaptations are under way in health care sectors, for example adaptive pharmaceuticals licensing advanced by Health Canada and the European Medicines Agency (*Oye, 2012*), avoiding a binary acceptance or rejection of a specific product or technology in favour of "stepwise learning under conditions of acknowledged uncertainty, with initial limits on use, iterative phases of data gathering and regulatory evaluation" (*Oye, 2012* p.22).

Many countries and industry sectors now have policies promoting this approach – for example, the EU Principle of Proportionality, the UK accelerated access review, and the OECD recommendations on regulatory policy and governance, potentially marking a sea-change in the regulatory mind-set, and allowing more timely access to incremental types of innovation while still exploring the safe development of more disruptive innovation (*Tait and Banda, 2016*). In the EU, similar approaches may be applied to current GM regulation and to future regulation of biological engineering, for example, by favouring products and processes that can be monitored, 'recalled' or 'reversed'. However, so far there has been little movement in this direction, particularly for applications with potential environmental impacts, or application in food and farming industries. The reversibility of a given genetic technology will depend on its interplay with biological, ecological and social environments, and its promise does not necessarily mean it provides the best option to address the targeted challenge; as we highlighted in 'New approaches to synthetic gene drives' above.

### Some comments on the process

There are a number of caveats and considerations in the approach we have taken. The first concerns the Delphi technique, on which we based our structured elicitation. Originally developed for quantitative forecasting, the Delphi technique has a mixed track record. Its critics argue that it confuses opinion with systematic prediction, produces false precision, and imbues the result with undue confidence (*Sackman, 1975*). If forecasts of precisely defined events are sought (and if past data is available), tools such as trend analysis will likely give more accurate predictions, and could be used in conjunction with the Delphi method. We seek only to structure qualitative group judgments about a broad range of complex futures, for which we do not have neat datasets to extrapolate.

Compared with other elicitation approaches, such as traditional meetings, the Delphi method has also been found to *improve* forecasts and group judgments (*Rowe and Wright, 2001*). We believe the method's benefits transfer to broader foresight contexts.

Another feature of any group elicitation is that diverging opinions of individual contributors can be masked in aggregated scores. Our analysis of first round data indicates that while there was considerable diversity in the raw scores provided by individual participants, the inter-rater concordance was substantial and statistically highly significant (Kendall's $W=0.150$, p-value $< 10^{-15}$). The rank correlation between individual participants ranged between 0.002 and 0.463, with a median of 0.112 (Spearman's rho). Forty-eight of the 210 inter-rater correlations (23%) were statistically significant (Spearman correlation p-value $< 0.05$), further indicating that while the participants represented a wide variety of viewpoints, there was a core of shared opinions. Those who do agree, agree quite strongly.

Related to this, we do acknowledge that the issues raised in this paper reflect the people involved in the process, which is why we explicitly encouraged contributors to seek ideas from beyond their immediate circles, and attempted to capture a broad array of perspectives and experience in the core participants. Nevertheless, the participants were all UK- or US-based, and a future scan of this kind would benefit from including contributors from other parts of the world, particularly China, a region at the forefront of bioengineering, and where unpublished or locally published research is relatively difficult to access. Furthermore, our scan more heavily reflected the views of scholars and innovators than those of industry (although many participants had insight into industry through their consultancies).

We did not include policy makers directly in our initial scan, as we wished to restrict the size and composition of the exercise to those at the exploratory end of research and innovation. Follow-up exercises could, however, involve government representatives to help identify the most actionable issues. Such an exercise could utilise an established framework for back-casting (*Holmberg and Robert, 2000*), or road mapping, or another process for assessing impact and urgency of the identified issues for their organisation (*Sutherland et al., 2012*). Bringing together a group of policy makers in a follow-up exercise also encourages the prioritisation of cross-organizational issues, setting an agenda for sharing knowledge and developing policy collaboratively. Ideally, feasibility assessments of the options available would be included (as carried out in the extension of the recent Antarctic scan (*Kennicutt et al., 2016*), outlined in 'Aims' above). The annual horizon scan of conservation issues has experienced no shortage of novel material. We suggest repeating the base scan at regular intervals, such as biennially.

The issues presented here are not intended to be exhaustive, nor reflective of what we think are the most *important* issues. We stress that a number of issues did not make the final list because they were a less appropriate fit with the aims of the paper, *not* because they were deemed less important. Specifically, they may have been considered (i) too well known or widely discussed in the bioengineering community already (e.g. extreme risks posed by a small group of people with increased access to resources and with malicious intent); (ii) too broad (e.g. adaptive governance as a stand-alone issue) and/or (iii) too improbable, scientifically challenging or far-off in the future (e.g. xenobiology, or engineering neural cells to better interface with computers). We do recognise that those in the latter category can make good horizon issues precisely for the reasons we excluded them, and a separate process could focus on this category of issues. But here, we present a set of issues that we believe are likely to emerge in the field of biological engineering over the coming years.

## Acknowledgements

This publication was made possible by the Centre for the Study of Existential Risk, the Synthetic Biology Strategic Research Initiative (both at the University of Cambridge), and the Future of Humanity Institute (University of Oxford). The workshop was co-funded by the Templeton World Charity Foundation (TWCF) and the European Research Council (ERC) under the European Union's Horizon 2020 research and innovation programme (grant agreement No 669751). We thank those outside the author list who suggested issues, in particular Shahar Avin and Lalitha Sundaram. The opinions expressed in this publication are those of the authors and do not necessarily reflect the views of the TWCF or ERC, and neither the TWCF or ERC are responsible for any use that may be made of the information it contains.

**Bonnie C Wintle** is in the Centre for the Study of Existential Risk, University of Cambridge, Cambridge, United Kingdom

bondiewints@gmail.com

http://orcid.org/0000-0003-0236-6906

**Christian R Boehm** is in the Max Planck Institute of Molecular Plant Physiology, Potsdam, Germany, and the Centre for the Study of Existential Risk, University of Cambridge, Cambridge, United Kingdom

cboehm@mpimp-golm.mpg.de

http://orcid.org/0000-0002-6633-7998

**Catherine Rhodes** is in the Centre for the Study of Existential Risk, University of Cambridge, Cambridge, United Kingdom

https://orcid.org/0000-0002-7747-2597

**Jennifer C Molloy** is in the Department of Plant Sciences, University of Cambridge, Cambridge, United Kingdom

https://orcid.org/0000-0003-3477-8462

**Piers Millett** is in the Future of Humanity Institute, University of Oxford, Oxford, United Kingdom

**Laura Adam** is in the Department of Electrical Engineering, University of Washington, Seattle, United States

**Rainer Breitling** is in the Manchester Synthetic Biology Research Centre (SYNBIOCHEM), Manchester Institute of Biotechnology, University of Manchester, Manchester, United Kingdom

http://orcid.org/0000-0001-7173-0922

**Rob Carlson** is at Bioeconomy Capital, Seattle, United States

**Rocco Casagrande** is at Gryphon Scientific, Takoma Park, United States

**Malcolm Dando** is in Division of Peace Studies and the Bradford Centre for International Development, University of Bradford, Bradford, United Kingdom

**Robert Doubleday** is in the Centre for Science and Policy, University of Cambridge, Cambridge, United Kingdom

**Eric Drexler** is in the Future of Humanity Institute, University of Oxford, Oxford, United Kingdom

https://orcid.org/0000-0002-7309-1738

**Brett Edwards** is in the Department of Politics, Languages & International Studies, University of Bath, Bath, United Kingdom

**Tom Ellis** is in the Centre for Synthetic Biology and Innovation, Imperial College London, London, United Kingdom

**Nicholas G Evans** is in the Department of Philosophy, University of Massachusetts, Lowell, United States

https://orcid.org/0000-0002-3330-0224

**Richard Hammond** is at Cambridge Consultants Limited, Cambridge, United Kingdom

**Jim Haseloff** is in the Department of Plant Sciences, University of Cambridge, Cambridge, United Kingdom

**Linda Kahl** is at the BioBricks Foundation, San Francisco, United States

http://orcid.org/0000-0003-4139-8505

**Todd Kuiken** is in the Genetic Engineering & Society Center, North Carolina State University, Raleigh, United States

https://orcid.org/0000-0001-7851-6232

**Benjamin R Lichman** is at the John Innes Centre, Norwich, United Kingdom

https://orcid.org/0000-0002-0033-1120

**Colette A Matthewman** is at the John Innes Centre, Norwich, United Kingdom

https://orcid.org/0000-0003-2351-6221

**Johnathan A. Napier** is at Rothamsted Research, Harpenden, United Kingdom

https://orcid.org/0000-0003-3580-3607

**Seán S ÓhÉigeartaigh** is in the Centre for the Study of Existential Risk, University of Cambridge, Cambridge, United Kingdom

**Nicola J Patron** is in the Earlham Institute, Norwich, United Kingdom

https://orcid.org/0000-0002-8389-1851

**Edward Perello** is at Desktop Genetics, London, United Kingdom

**Philip Shapira** is at the Manchester Institute of Innovation Research, Alliance Manchester Business School, University of Manchester, Manchester, United Kingdom, and the School of Public Policy, Georgia Institute of Technology, Atlanta, United States

http://orcid.org/0000-0003-2488-5985

**Joyce Tait** is at the Innogen Institute, University of Edinburgh, Edinburgh, United Kingdom

**Eriko Takano** is in the Manchester Synthetic Biology Research Centre (SYNBIOCHEM), Manchester Institute of Biotechnology, University of Manchester, Manchester, United Kingdom

http://orcid.org/0000-0002-6791-3256

**William J Sutherland** is in the Conservation Science Group, Department of Zoology, University of Cambridge, Cambridge, United Kingdom

*Author contributions:* Bonnie C Wintle, Conceptualization, Data curation, Formal analysis, Investigation, Visualization, Methodology, Writing—original draft, Project administration, Writing—review and editing; Christian R Boehm, Data curation, Formal analysis, Investigation, Writing—original draft, Writing—review and editing; Catherine Rhodes, Conceptualization, Investigation, Writing—original draft, Project administration, Writing—review and editing; Jennifer C Molloy, Conceptualization, Investigation, Writing—original draft, Writing—review and editing; Piers Millett, Conceptualization, Investigation, Writing—original draft; Laura Adam, Rocco Casagrande, Brett Edwards, Tom Ellis, Benjamin R Lichman, Colette A Matthewman, Johnathan A Napier, Nicola J Patron, Philip Shapira, Investigation, Writing—original draft, Writing—review and editing; Rainer Breitling, Formal analysis, Investigation, Methodology, Writing—original draft, Writing—review and editing; Rob Carlson, Malcolm Dando, Robert Doubleday, Nicholas G Evans, Richard

Hammond, Jim Haseloff, Linda Kahl, Todd Kuiken, Edward Perello, Joyce Tait, Eriko Takano, Investigation, Writing—original draft; Eric Drexler, Investigation, Methodology, Writing—original draft; Seán S ÓhÉigeartaigh, Conceptualization, Supervision, Funding acquisition, Investigation, Writing—original draft; William J Sutherland, Conceptualization, Supervision, Methodology, Project administration, Writing—review and editing, W.J.S. facilitated the workshop and designed the original methodology on which the process was based

**Competing interests:** The authors declare that no competing interests exist.

## Funding

| Funder | Grant reference number | Author |
|---|---|---|
| Templeton World Charity Foundation | TWCF0128/AB82 | Bonnie C Wintle Catherine Rhodes Seán S ÓhÉigeartaigh |
| Gates Cambridge Trust | | Christian R Boehm |
| Biotechnology and Biological Sciences Research Council | Manchester Synthetic Biology Research Centre for Fine and Speciality Chemicals BB/M017702/1 | Rainer Breitling Philip Shapira Eriko Takano |
| Biotechnology and Biological Sciences Research Council | Institute Strategic Programme Grant BBS/E/C/00005207 | Johnathan A. Napier |

The funders had no role in study design, data collection and interpretation, or the decision to submit the work for publication.

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
