## [Decision Letter]

Thank you for submitting your manuscript "A 2017 horizon scan of emerging issues in biological engineering" to *eLife* for consideration as a Feature Article. Your article has been reviewed by three peer reviewers, and the evaluation has been overseen by myself (the *eLife* Features Editor). The following individuals involved in review of your submission have agreed to reveal their identity: Ian Lipkin (Reviewer #2); Jeantine Lunshof (Reviewer #3).

We invite you to revise your manuscript to address the points listed below.

Summary:

The piece is well written and identifies many issues that need to receive more attention. The strength of the project – and of the article – lies in the method and in the way the procedure was applied to the very wide horizon of the study topic of developments in biological engineering. For people active in the field, not many surprises, but many of these issues are not well known outside the synthetic biology realm.

Essential revisions:

1) An inherent limitation of the project/article is the lack of diversity among the experts in terms of geography and worldview. While geography and worldview do not necessarily overlap, it seems to be the case in this group and in this project. In the "issues" addressed, any reference to the role that cultural and value aspects may play in the design and the uptake of biological engineering technologies and applications is lacking.

From the outset, there is a strong focus on current development and future need of regulation. However, the discussion of regulation is mostly focused on the EU/UK/US, even though other countries in the world are already key players in genomic sciences and biological engineering.

Please be more explicit about the lack of diversity in expert input near the start of the article.

Please consider changing the title of the article to reflect the geographical limitations of the project and its conclusions.

Please consider moving the Box "Emerging Issues Regulation" to the main text because placing this material (which is mostly about the EU/UK/US) in a separate box lends it additional emphasis.

2) A major challenge for the authors is to ensure that this work receives the attention it deserves. As such the manuscript would benefit from extra discussions about priorities and timings.

Priorities. The article states that the authors used an "iterative process to prioritise 20 issues." Are all 20 issues of equal priority? The issues identified have differing impacts, timings, and implications for research and public policy, but they are presented as a group with no obvious prioritization. Run this thought experiment: The head of NSF or the EPSRC/BBSRC decides to invest an additional $50 million in synthetic biology over a ten year period. You have five minutes to argue how to best distribute the funds based on your horizon scan. What do you tell them?

Timing. Relevancy and timing are key to impact. The article talks about the identifying issues with potential impact on the "medium-term future" (15-20 years). That is a laudable goal, but difficult to achieve in complex socio-technical systems. Looking back at the field of synthetic biology might give us some idea of how difficult it will be to look ahead 15-20 years. 15 years ago the first MIT synthetic biology course had not yet occurred (November 2003) nor had SB1.0.

Many of the 20 issues will have impact much sooner or are already having impacts (like the significant increase in defense spending for synthetic biology research). The use of gene drives for plants and animals already motivated (forced) the US FDA to issue two guidance documents in January 2017 (one is now in OMB review).

One way to get the attention of decision makers is to present a mix of near terms (<5 years), medium, and long-term issues. Few people can think out 15-20 years due to cognitive biases, like anchoring or hindsight effects. Added to these biases are institutional and learning failures. It seems that many of the 20 issues could be further disaggregated in terms of timing to provide a smaller sub-group of on-the-immediate-horizon issues that could be leveraged to get the attention of decision-makers.

Please include a discussion that makes clear which of the 20 issues are, in your opinion, of the highest priority and/or in need of the most attention now.

Related to this, please consider including a table showing the original 70 issues, along with some type of rank ordering (and clearly indicating the 20 issues that are discussed in the article).

3) The hope behind many foresight exercises is that someone with budgetary or policy power takes note. That means that the demand side is as important, or more important, than supply.

Who in this group of authors represented the demand side and how was the research informed by possible end users and their needs?

4) The research design of a horizon scanning exercise should address organizational approaches, communications strategies, and boundary-spanning techniques to help ensure their use. Creating actionable results is understandably very difficult. Even organizations which had direct linkages to governmental decision-making process, like the former Danish Technology Board or US Office of Technology Assessment, had problems achieving impacts with these types of studies. One-shot horizon scans in rapidly changing systems need to be repeated and modified as needed – an adaptive learning approach. The paper acknowledges the need to iterate, but does not outline how, who, or at what frequency.

Please say more about what needs to happen next. For instance, would the authors recommend repeating the workshop each year? Should it be expanded to include representative from China or other countries? Should more policy makers be included in the next exercise? What needs to change that can inform other exercises like this?

5) A number of weaknesses are apparent in the "issues" sections. The quality of the reported case studies is very uneven. While many are clear, systematic, well structured and with up-to-date factual information, a number does not meet the standards.

Regenerative medicine: The description is imprecise, the text is crowded, an abundance of speculative statements.

Human genome augmentation: Imprecise, unclear terminology (including the term 'human genome augmentation' itself). Tendentious and unclear use of language: "… balance a desire to maintain healthy and competitive human capital with the risk of creating novel sociogenetic pecking orders."

Tendentious language and neologisms are not very useful in the context of suggestions for regulation.

Producing vaccines and human therapies in plants: The presentation lacks clarity and the many presented 'items' lack coherence. A concrete regulation relevant conclusion is missing.

Biology as an information science: Impacts on global governance – lack of clarity. One example: ".… the most valuable biological resources are collections of intangible assets that can be relocated online."

Intersection of information security and bio-automation: Unclear reasoning, lack of coherence.

Low-profile biocrime: There are too many and to divergent examples and details; a clear direction is lacking.

6) Please include one or more sentences that clearly define "biological engineering" at the start of the article.

7) The explanation/discussion of the granularity model needs to be clearer.

8) The modified Delphi technique used is a very blunt foresight instrument and the authors should have been more open about its downsides. For instance, the RAND corporation, who developed the Delphi technique, undertook a critical examination of more than a decade of its exercises (Sackman, 1975). The findings are still applicable. The Delphi exercises had a tendency to be highly vulnerable to the concept of "expert," seriously confused aggregations of raw opinion with systematic prediction, typically generated snap answers to ambiguous questions (representing inkblots of the future), and gave an exaggerated illusion of precision, misleading uninformed users of results. This doesn't mean that the technique should not be used, but applied with caution and in combination with other forecasting approaches, for instance, trend analyses.

Please include a brief discussion of the shortcomings of the Delphi technique.

---

## [Author Response]

Summary:The piece is well written and identifies many issues that need to receive more attention. The strength of the project – and of the article – lies in the method and in the way the procedure was applied to the very wide horizon of the study topic of developments in biological engineering. For people active in the field, not many surprises, but many of these issues are not well known outside the synthetic biology realm.

We thank the editor and reviewers for providing such helpful and reasonable feedback.

Essential revisions:1) An inherent limitation of the project/article is the lack of diversity among the experts in terms of geography and worldview. While geography and worldview do not necessarily overlap, it seems to be the case in this group and in this project. In the "issues" addressed, any reference to the role that cultural and value aspects may play in the design and the uptake of biological engineering technologies and applications is lacking.From the outset, there is a strong focus on current development and future need of regulation. However, the discussion of regulation is mostly focused on the EU/UK/US, even though other countries in the world are already key players in genomic sciences and biological engineering.Please be more explicit about the lack of diversity in expert input near the start of the article.

Added to 'Aims' section: "Although we have attempted to capture an assortment of backgrounds, expertise, agendas, and demographics (including age, gender and career stage), we acknowledge that we present perspectives of researchers based in the UK and US."

Please consider changing the title of the article to reflect the geographical limitations of the project and its conclusions.

Changed title to " A transatlantic perspective on 20 emerging issues in biological engineering ".

Please consider moving the Box "Emerging Issues Regulation" to the main text because placing this material (which is mostly about the EU/UK/US) in a separate box lends it additional emphasis.

We have addressed this as outlined below.

2) A major challenge for the authors is to ensure that this work receives the attention it deserves. As such the manuscript would benefit from extra discussions about priorities and timings.Priorities. The article states that the authors used an "iterative process to prioritise 20 issues." Are all 20 issues of equal priority? The issues identified have differing impacts, timings, and implications for research and public policy, but they are presented as a group with no obvious prioritization. Run this thought experiment: The head of NSF or the EPSRC/BBSRC decides to invest an additional $50 million in synthetic biology over a ten year period. You have five minutes to argue how to best distribute the funds based on your horizon scan. What do you tell them?

We have organised the issues in terms of [policy] relevance under different time horizons (as also suggested by the reviewer, see 'Timing' below). The prioritisation, then, depends on the focus of the policy maker.

Please also see added text for justification of our decision not to prioritise the issues according to rank order. Added to 'Discussion section': "Having completed the iterative process of culling low-scoring issues as described above, we found little separating the top 20 issues. To avoid over-emphasizing slight differences in score, we chose not to present the top 20 issues in rank order. The complexity of the field requires a comprehensive, multi-faceted approach."

Further to this, the differences in rank between topics that are ranked close to each other are never statistically significant, and no topic received a statistically better score than the 6 topics ranked below it (Wilcoxon rank sum test, p-value > 0.10). While this does not mean that the ranking is uninformative (e.g., the most highly-ranked original topic after Round 1, "Platform technologies to address emerging disease pandemics", clearly stands out as particularly relevant, receiving significantly better scores than 61 of the initial 70 topics, with a Wilcoxon p-value < 0.10, while each of the top 20 topics, has a significantly better ranking than each of the 23 lowest scoring topics), it strongly emphasises that individual differences in position would be liable to over-interpretation.

Another problem with presenting in rank order is that, in our experience with horizon scanning, broader issues tend to rank higher than more specific issues (as discussed in the 'granularity' paragraph). But broader issues are not necessarily higher priorities.

Timing. Relevancy and timing are key to impact. The article talks about the identifying issues with potential impact on the "medium-term future" (15-20 years). That is a laudable goal, but difficult to achieve in complex socio-technical systems. Looking back at the field of synthetic biology might give us some idea of how difficult it will be to look ahead 15-20 years. 15 years ago the first MIT synthetic biology course had not yet occurred (November 2003) nor had SB1.0.Many of the 20 issues will have impact much sooner or are already having impacts (like the significant increase in defense spending for synthetic biology research). The use of gene drives for plants and animals already motivated (forced) the US FDA to issue two guidance documents in January 2017 (one is now in OMB review).One way to get the attention of decision makers is to present a mix of near terms (<5 years), medium, and long-term issues. Few people can think out 15-20 years due to cognitive biases, like anchoring or hindsight effects. Added to these biases are institutional and learning failures. It seems that many of the 20 issues could be further disaggregated in terms of timing to provide a smaller sub-group of on-the-immediate-horizon issues that could be leveraged to get the attention of decision-makers.

We have roughly grouped issues by impact according to time horizon. We believe that grouping the issues in this way is a reasonable alternative to presenting issues by rank priority.

Added to 'Procedure’ section: "The final list of issues is reported below, roughly grouped according to their relevance in the near (< 5 years), intermediate (5-10 years), and longer (> 10 years) terms. 'Relevance' is a subjective and fuzzy measure and could be determined by any number of factors, such as whether the issue is considered underway, established, or at a tipping point by the indicated year. Applying criteria such as these is difficult for issues that are composed of multiple elements. So here, 'relevance' refers to how directly, and by extension, how soon the overall issue could measurably interact with risks to society."

Also, added to 'Discussion' section: "Some policy- or decision-making bodies may focus on preparing for distant futures, and on long term issues that may otherwise be overshadowed by current, more pressing priorities. Others may focus on nearer term issues that require immediate attention. Large science funding bodies tend to consider a diverse suite of issues spread across a range of time horizons. To put the top 20 issues in a temporal context, we roughly grouped them according to their relevance in the near (< 5 years), intermediate (5-10 years), and longer (> 10 years) term. Applications and research with the potential for near-term impacts on critical systems, such as global food and fuel supply, ecosystems, health and geopolitics, thus appear in the first category. Those that influence society indirectly, via platforms, ownership models, markets or future infrastructure, may have less immediate societal impact and thus appear in the latter category."

Please include a discussion that makes clear which of the 20 issues are, in your opinion, of the highest priority and/or in need of the most attention now.

As noted above, we have grouped by time horizon, which, depending on the remit of the organisation, could be used to further prioritise issues.

Related to this, please consider including a table showing the original 70 issues, along with some type of rank ordering (and clearly indicating the 20 issues that are discussed in the article).

Please see justification above for not presenting issues in rank order. Also, given the title changes and reworking of some issues, we do not think adding the original issues will be that useful. If this is deemed essential for publication, we can consider adding to Supplementary information.

3) The hope behind many foresight exercises is that someone with budgetary or policy power takes note. That means that the demand side is as important, or more important, than supply.Who in this group of authors represented the demand side and how was the research informed by possible end users and their needs?

This is an important point, and relates closely to the one below it. We therefore address these two points together.

4) The research design of a horizon scanning exercise should address organizational approaches, communications strategies, and boundary-spanning techniques to help ensure their use. Creating actionable results is understandably very difficult. Even organizations which had direct linkages to governmental decision-making process, like the former Danish Technology Board or US Office of Technology Assessment, had problems achieving impacts with these types of studies. One-shot horizon scans in rapidly changing systems need to be repeated and modified as needed – an adaptive learning approach. The paper acknowledges the need to iterate, but does not outline how, who, or at what frequency.Please say more about what needs to happen next. For instance, would the authors recommend repeating the workshop each year? Should it be expanded to include representative from China or other countries? Should more policy makers be included in the next exercise? What needs to change that can inform other exercises like this?

Added text to 'Discussion' section: "There are a number of caveats and considerations in the approach we have taken. […]…a future scan of this kind would benefit from including contributors from other parts of the world, particularly China, a region at the forefront of bioengineering, and where unpublished or locally published research is relatively difficult to access […] We did not include policy makers directly in our initial scan, as we wished to restrict the size and composition of the exercise to those at the exploratory end of research and innovation. Follow up exercises could, however, involve representatives from government to help identify the most actionable issues. Such an exercise could utilise an established framework for back-casting [43], or road mapping, or another process for assessing impact and urgency of the identified issues for their organisation [107]. Bringing together a group of policy makers in a follow up exercise also encourages the prioritisation of cross-organizational issues, setting an agenda for sharing knowledge and developing policy collaboratively. Ideally, feasibility assessments of the options available would be included (as carried out in the extension of the recent Antarctic scan [54], outlined in 'Aims' above). The annual horizon scan of conservation issues has experienced no shortage of novel material. We suggest repeating the base scan at regular intervals, such as biennially."

5) A number of weaknesses are apparent in the "issues" sections. The quality of the reported case studies is very uneven. While many are clear, systematic, well structured and with up-to-date factual information, a number does not meet the standards.Regenerative medicine: The description is imprecise, the text is crowded, an abundance of speculative statements.

Please see manuscript for new issue summary. Among other edits, we have changed the title and focus to medicinal applications only (not augmentation).

Human genome augmentation: Imprecise, unclear terminology (including the term 'human genome augmentation' itself). Tendentious and unclear use of language: "… balance a desire to maintain healthy and competitive human capital with the risk of creating novel sociogenetic pecking orders."Tendentious language and neologisms are not very useful in the context of suggestions for regulation.

Please see manuscript for new issue summary. Among other edits, we have changed the title and focus to medicinal applications only (not augmentation).

Producing vaccines and human therapies in plants: The presentation lacks clarity and the many presented 'items' lack coherence. A concrete regulation relevant conclusion is missing.

Please see manuscript for new issue summary. Among other edits, we added the following conclusion: "The 2012 approval of Elelyso (Protalix) for commercial use in humans to treat Gaucher’s disease has paved the way [70] for a number of therapeutics and vaccines targeting conditions ranging from influenza to non-Hodgkins lymphoma [44]. This widened scope and accumulation of examples and successes signals a shift toward rapidly deployed, industrial scale plant-based production of new therapies for emerging diseases, which will require an equally responsive regulatory landscape for testing and deployment."

Biology as an information science: Impacts on global governance – lack of clarity. One example: ".… the most valuable biological resources are collections of intangible assets that can be relocated online."

Please see manuscript for new, clearer issue summary.

Intersection of information security and bio-automation: Unclear reasoning, lack of coherence.

Please see manuscript for new, clearer issue summary.

Low-profile biocrime: There are too many and to divergent examples and details; a clear direction is lacking.

Please see manuscript for new issue summary.Refocused on 'Corporate espionage and biocrime', and renamed accordingly. Divergent examples removed.

6) Please include one or more sentences that clearly define "biological engineering" at the start of the article.

Added definition to 'Aims' section: "Biological engineering can be understood as the application of principles and techniques from engineering to biological systems, often with the articulated goal of addressing 'real-world' problems"

7) The explanation/discussion of the granularity model needs to be clearer.

Please see added text. Also, the text on the actual example has been cut back (changes tracked).

Added text to 'Procedure' section: "For submitted issues to be comparable with each other, they need to be framed at similar levels of granularity. Issues that are very broad, such as 'regulation of bioengineering', will encompass a whole suite of more detailed issues, so typically score higher than a single, highly specific issue. But these really broad topics rarely make good horizon scanning issues, as they tend to be already well known, and are too vague to act on. To help ensure that issues were submitted with an appropriate level of granularity, an example topic was circulated that was framed at five different scales (Figure 2)."

Note to Editor: Now that we have clarified granularity, we could delete the whole 'dual use' example and figure, if it takes too many tangential words to explain. Please advise on your preference.

8) The modified Delphi technique used is a very blunt foresight instrument and the authors should have been more open about its downsides. For instance, the RAND corporation, who developed the Delphi technique, undertook a critical examination of more than a decade of its exercises (Sackman, 1975). The findings are still applicable. The Delphi exercises had a tendency to be highly vulnerable to the concept of "expert," seriously confused aggregations of raw opinion with systematic prediction, typically generated snap answers to ambiguous questions (representing inkblots of the future), and gave an exaggerated illusion of precision, misleading uninformed users of results. This doesn't mean that the technique should not be used, but applied with caution and in combination with other forecasting approaches, for instance, trend analyses.Please include a brief discussion of the shortcomings of the Delphi technique.

Created a new Discussion sub-section called "Some comments on the process" and added the following text: "There are a number of caveats and considerations in the approach we have taken. The first concerns the Delphi technique, on which we based our structured elicitation. Originally developed for quantitative forecasting, the Delphi technique has a mixed track record. Its critics argue that it confuses opinion with systematic prediction, produces false precision, and imbues the result with undue confidence [97]. If forecasts of precisely defined events are sought (and if past data is available), tools such as trend analysis will likely give more accurate predictions, and could be used in conjunction with the Delphi method. We seek only to structure qualitative group judgements about a broad range of complex futures, for which we do not have neat datasets to extrapolate. Compared with other elicitation approaches, such as traditional meetings, the Delphi method has also been found to improve forecasts and group judgements [95]. We believe the method's benefits transfer to broader foresight contexts."